# From Seed to Seedling: Influence of Seed Geographic Provenance and Germination Treatments on Reproductive Material Represented by Seedlings of *Robinia pseudoacacia*

Andreea M. Roman [1,2], Alina M. Truta [1,*], Irina M. Morar [1,*], Oana Viman [1], Catalina Dan [1], Adriana F. Sestras [1], Liviu Holonec [1], Monica Boscaiu [3] and Radu E. Sestras [1]

[1] Faculty of Horticulture, University of Agricultural Sciences and Veterinary Medicine, 3-5 Mănăștur Street, 400372 Cluj-Napoca, Romania; andreearoman34@gmail.com (A.M.R.); oana.viman@usamvcluj.ro (O.V.); catalina.dan@usamvcluj.ro (C.D.); adriana.sestras@usamvcluj.ro (A.F.S.); lholonec@usamvcluj.ro (L.H.); rsestras@usamvcluj.ro (R.E.S.)

[2] Forestry College Transilvania, 1 Gării Street, 425200 Năsăud, Romania

[3] Mediterranean Agroforestry Institute (IAM), Universitat Politècnica de Valencia, Camino de Vera s/n, 46022 Valencia, Spain; mobosnea@eaf.upv.es

[*] Correspondence: alina.truta@usamvcluj.ro (A.M.T.); irina.todea@usamvcluj.ro (I.M.M.)

**Abstract:** The influence of the geographical origin of eight Romanian provenances of *Robinia pseudoacacia* on the characteristics of seeds, germination, and growth of seedlings in young stages of life was analyzed. Four experiments were undertaken to test seed germination (thermal treatment at distinct temperatures, mechanical scarification, acetone 90%, and biostimulator). The germination percentage showed that scarification treatment provided the best results among all treatments (41.7%). Seeds soaked in water at 100 °C provided the second-highest germination rate. Furthermore, the same two treatments also assured the highest values for the seedlings' length. There were registered significant differences among the provenances for the analyzed characteristics, the seed germination capacity, and the growth rate of the seedlings in the first years of life. The study highlighted the resources that could ensure good quality of the reproductive forest material, which can be used in new afforestation and breeding programs. Pearson correlations and multivariate analysis provided interesting and useful information about *R. pseudoacacia* provenances and 13 characteristics of the seeds and seedlings, highlighting the relationship among them. The results could be of interest for the efficient use of forest genetic resources and the obtention of quality reproductive material in black locust.

**Keywords:** black locust; seed origin; growth; germination indices

## 1. Introduction

The black locust (*Robinia pseudoacacia* L., family Fabaceae) is a well-known forest species native to the eastern part of North America, from where it has been widely introduced throughout the world [1]. The species was introduced in Europe at the beginning of the 17th century, and now has a significant distribution, from sub-Mediterranean to warm continental climates [2,3]. Black locust is a species with acknowledged attributes. Ecologically, it has the ability to quickly colonize a site that is in danger of eroding, and as an early pioneer species, it leaves the site in a state that can be further occupied by other late-succession species [4]. In regard to its utility, the species has an accelerated growth, thus producing useful biomass quickly. It is hence widely planted to control soil erosion, particularly in strip-mined and sandy regions; used within shelterbelts and for ornamental purposes; planted for forestry restoration; and appreciated as a melliferous and nitrogen-fixing species, and the wood has several practical uses [5].

Black locust reproduces sexually and asexually; it can propagate by seeds, spring root cuttings, or dormant sucker divisions. For breeding studies, generative reproduction is

taken into consideration: It produces genetic variation in the offspring generations (exploitation of variability in such a way), the seedlings have a high survival rate, and the species can easily adapt to new environments [2,6]. In its native and introduced ranges, natural regeneration of black locust is primarily vegetative; however, the species also regulates its spread on sexual reproduction, with bee-pollinated hermaphrodite flowers and prolific seed production and dispersal [7,8]. Seed ripening takes place from late summer until early autumn, and pods are shed from September to April. Consequently, *R. pseudoacacia* forms short-term aerial seed banks [9], as well as long-term soil seed banks [10,11]. The success or failure of generative regeneration generally depends on the seeds' ability to germinate under different environmental conditions such as temperature, water availability, and light [12]. These factors may have the farthest values, so the impact upon all physiological processes and their effects can be essential in both the survival and the establishment of seedlings [11,13,14].

Seeds of black locust are characterized by exogenous dormancy due to the structure of the seed tegument, which is impermeable to water. For this reason, artificial stratification, as well as thermic or chemical treatments, are necessary to apply. In this manner, the imbibition and course of germinative processes may be influenced and the dormancy stage inhibited. In the absence of germination treatments, the black locust seed coat can be damaged by either low (frosts) or high (including fires) temperatures, or by the activity of soil microflora. For economic and research reasons, several methods are currently used [15]: mechanical scarification, as well as chemical and thermal procedures to increase the germination capacity. Thus, each seed lot is recommended to be tested separately to determine the proper length of time for soaking methods and type of blends to be used and their concentration, due to the variability of seed coat structure and permeability [4]. Different solutions, concentrations, and durations of the methods used to break black locust seed dormancy and test seed germination are found in the specialized literature.

Germination represents a qualitative developmental reaction of seeds, a process that occurs at a specific point in time and plant life cycle within distinct conditions, but individual seeds may respond differently [16]. This leads to a situation in which the final germination percentage alone is not always sufficient for reporting concluding results for one species, due to the lack of ability to compare different sets of data (e.g., one lot of seeds may have germinated well before the other, but both attained the same final germination percentage).

As *R. pseudoacacia* has multiple valuable uses, but also undesirable features, the species behavior is worth studying under varying environmental factors to estimate the geographical area potentially invaded by *R. pseudoacacia* in the perspective of planning its management and biodiversity regulation [17]. Correlated with black locust utility, the frequent use nowadays and its invasive spread, all stages starting with seed harvest to germination and further tree growth, must be considered. Regeneration from seeds is now considered an important determinant in the successful invasion or spread pattern of black locust [9], and the role of abiotic factors is essential to understanding the germination responses of dispersed seeds [18]. Moreover, in Europe, information on seed regeneration is insufficient for accurate species management, and germination requirements are not completely clarified [11]. In this regard, the studies that try to assemble such distinct factors (i.e., the origin of seeds and seeds features, germination treatments, seedling growth) to gain a general overview of the process of producing valuable reproductive material for the species are welcome [6]. Forest reproductive material (FRM) as a notion is generally considered to be a generic name for seeds, cones, cuttings, and planting material (planting stock) used in forest establishment. Seed vigor is defined as the sum of those traits that determine the level of activity and performance of the seed during germination and seedling emergence [19]. There is no universally accepted vigor test for all seeds, but the following should be taken into account: the reproducibility of the chosen methods and the relationship between seed test results and seedling emergence.

All the above data justify any new approaches that would contribute to ensuring a valuable biological reproductive material of the species. The current study aims to provide an exhaustive image for the species, starting with the harvest of seeds from separate locations, testing germination based on different treatments, interpreting the data about germination with the most appropriate formulas from the specialized literature, as well as correlating all data obtained with the growth of *R. pseudoacacia* seedlings, in order to investigate the possible relationships established.

## 2. Materials and Methods

### 2.1. Biological Material

Seeds of *R. pseudoacacia* belonging to eight Romanian provenances (Figure 1) were analyzed: (1) Bistrița-Năsăud (RNP. Romsilva OSE Lechința UP.7, u.a.80C, coordinates 47°00′ N/24°20′ E), (2) Galați (RNP. Romsilva OS. Tecuci UP.VI, u.a.49B, coordinates 45°46′ N/27°30′ E), (3) Iași (RNP. Romsilva OS. Iași UP. III, u.a.105E, coordinates 47°25′ N/27°20′ E), (4) Satu-Mare (OS. Carei, UP III, u.a.57L, 58A, coordinates 47°42′ N/22°24′ E, (5) Botoșani (RNP. Romsilva OS. Darabani UP. III, u.a.26D, coordinates 48°02′ N/27°01′ E), (6) Arad (Private Orchard Bărzani (Ruben Budău), coordinates 46°19′ N/21°40′ E), (7) Râmnicu Vâlcea (RNP. Romsilva OS. Stoiceni UPII, u.a.5L, 5M, coordinates 45°03′ N/24°26′ E), and (8) Bihor (RNP. Romsilva. OS. Săcuieni UP. IV, u.a.45C, coordinates 47°32′ N/22°09′ E). All provenances represent forest stands attested by the Romanian Gene Reserved Forests and Seed Stands, included in the National Catalog of Forest Genetic Resources and Forest Reproductive Materials [20]. The seeds used for the current investigation were procured from official providers from respective Romanian seed source reservations.

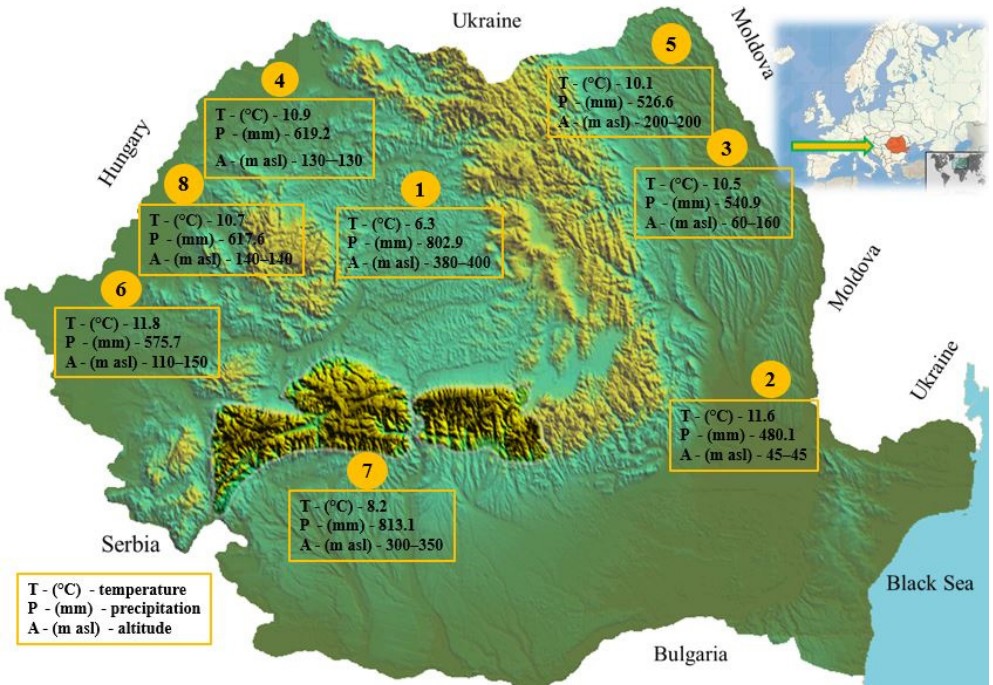

**Figure 1.** The Romanian provenances of *R. pseudoacacia* seeds.

The provenances for *R. pseudoacacia* are distinguished by different ecological parameters, such as average yearly temperature (°C), average annual precipitation (mm), and altitude (m asl) (Figure 1). The origin of the seeds was considered as it could influence germination and further seedling development, and thus was part of the current investigation. A visual sorting of *R. pseudoacacia* seeds was performed using the exterior condition as a criterion, so only seeds without visible damage were kept for the study. Afterwards, the seeds were deposited in well-ventilated conditions (inside temperature 20 °C) for approxi-

mately 6 months. Before field sowing, several measurements of the main phenotypic traits were performed for the seeds: length, width and thickness (mm), and weight (g), to be further correlated with the germination process. In August, in the year of sowing, the first measurements were made upon the *R. pseudoacacia* seedlings obtained: height and diameter (cm), presented as a mean for each provenance. After the seedlings' growth, in the fourth year of life, it was possible to additionally determine the number of ramifications—the number of branches on the stem (axis) of the plants.

## 2.2. Germination Stimulation

An important aspect of the study was represented by the investigation of the germination process, using different treatments and taking repeated measurements. *R. pseudoacacia* germination was investigated with 200 seeds/treatment (50 seeds × 4 repetitions) in Petri dishes with filtered paper at the bottom. Four treatments were applied. The first treatment referred to the mechanical scarification method, using abrasive sandpaper until the tegument was visibly damaged [17]. The second type of treatment (thermal treatment) involved water-soaked seeds at different temperatures [3,6]; seeds were soaked for 10 min in water at a temperature of 100 °C, 70 °C, and 40 °C. For the third treatment, seeds were soaked in biostimulator solution (Foliplant, 0.9%) for 10 min [21]. The fourth treatment used acetone (90%) solution for 10 min of seed imbibition [22].

All seeds were watered with distilled water prior to treatment. The measurements were taken daily for 14 days, starting with the second day after the germination treatments were instituted. For control, seeds were only watered when needed with distilled water to keep moist during the investigation period.

## 2.3. The Studied Germination Parameters

In order to have an exhaustive image of the germination process, the following indices were calculated using the formulas and methodology from the literature:

- Germination percentage, GP (%),

  GP = $\frac{Number\ of\ seeds\ germinated\ per\ day}{Total\ number\ of\ seeds\ placed\ to\ germination}$ × 100 [23–25].

- Germination index, GI,

  GI = $\frac{Number\ of\ germinated\ seeds}{Days\ from\ the\ first\ control}$ + ... + $\frac{Number\ of\ germinated\ seeds}{Days\ from\ the\ last\ control}$ [16,26].
  Determinations were made for 15 days in total

- Speed of emergence, SE (using germination speed/germinative energy),

  SE = $\frac{Number\ of\ germination\ seeds\ in\ the\ first\ day\ of\ germination}{Number\ of\ germinated\ seeds\ in\ the\ last\ day\ of\ germination}$ × 100 [27].

- Coefficient of germination speed, CRG,

- CRG = $[(n_1 + n_2 + n_n)/((n_1 \times T_1) + (n_2 \times T_2) + (n_3 \times T_3) + ... )]$ × 100 [28,29].
  $n_1$ = number of seeds germinated on day 1 ($T_1$); $n_2$ = number of seeds germinated on day 2 ($T_2$); $n_n$ = number of seeds germinated on day $n$ ($T_n$)

- Seedling vigor index, SVI,

  SVI = $\frac{Seedling\ length,\ in\ mm \times Germination\ percent}{100}$ [23,30].

## 2.4. Statistical Design and Data Analysis

The registered data for the seeds and seedlings were processed as the mean of traits and standard error of the means (SEM). The graphical representation of the relations between seed germination and the main traits of the seeds was made using bivariate kernel density estimation using Wessa software [31]. Analysis of variance (one-way ANOVA) was applied to the analyzed traits and then the Duncan test ($p < 0.05$) was used as a post hoc test for the analysis of differences. Among the general class of multiple comparisons procedures, the most suitable for the obtained data and sets of mean comparisons was considered to be Duncan's multiple range test (Duncan's MRT, $p < 0.05$) [32]. Pearson correlations among the analyzed characteristics were computed and linear regression was performed considering seed germination in terms of the number of days, or germination

and the main traits of the seeds. The data were subjected to a multivariate method, namely, canonical correspondence analysis (CCA). A multivariate principal component analysis graph for the eight provenances of *R. pseudoacacia* was created using Past software; the same software was used for the construction of a dendrogram as Euclidean distances among provenances and investigated traits.

## 3. Results

### 3.1. Seed Characteristics

Seed traits such as thickness, length, width, and weight are represented as a boxplot diagram (Figure 2), which consists of the minimum and maximum range values, the upper and lower quartiles, as well as the median, to summarize the distribution of the dataset recorded. The minimum seed length registered was 3.2 mm in the provenance of Botoșani and the maximum was 6.4 mm in Arad. The minimum seed width was also recorded in Botoșani (1.9 mm), but the highest value was in Vâlcea (3.9 mm). The seed thickness ranged between 1.2 mm (Botoșani) and 2.2 mm (Galați), whereas the seed weights were noted between 0.051 g (Bihor) and 0.061 (Iași) (Figure 2). Seeds from Bihor provenance had a relatively low variability for all seed traits. The mean of empirical data was close to the median in the box plot for seed length and thickness for Vâlcea and Galați. Hence, Arad and Vâlcea provenances stand out for superior seed characteristics.

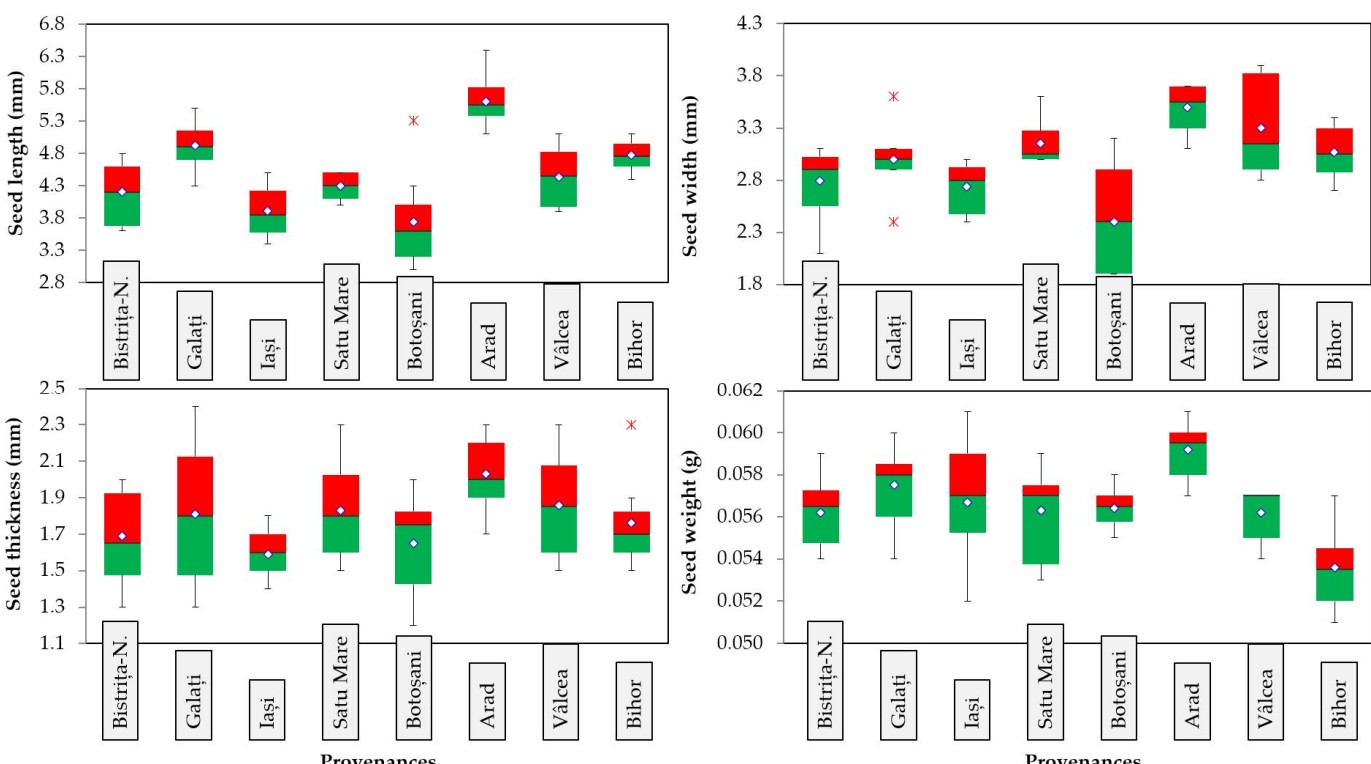

**Figure 2.** The principal traits of the seeds, based on the provenances of *R. pseudoacacia*. Each box plot represents minimum, lower quartile, median, upper quartile, and maximum values; in addition, the mean of the data is symbolized with a rhombus in the boxplot.

The main traits of the seeds, based on the provenances of *R. pseudoacacia*, are illustrated using the mean values (Figure 3). Once again, Arad provenance is worth noting, where all the traits analyzed were recorded with superior and significant mean values compared with the other provenances. Furthermore, the minimum seed characteristics were recorded differently in the investigated provenances, except in Botoșani, which registered the minimum values for length (3.7 mm) and width (2.4 mm). The mean values, simplistic as it is, show the tendency of the data for each provenance and are useful to note, even though the

outliers were quite distant, especially for the weight of *R. pseudoacacia* seeds investigated (0.054 g (Bihor) and 0.059 g (Arad)). It will be further investigated how these data correlate with germination and seedlings growth.

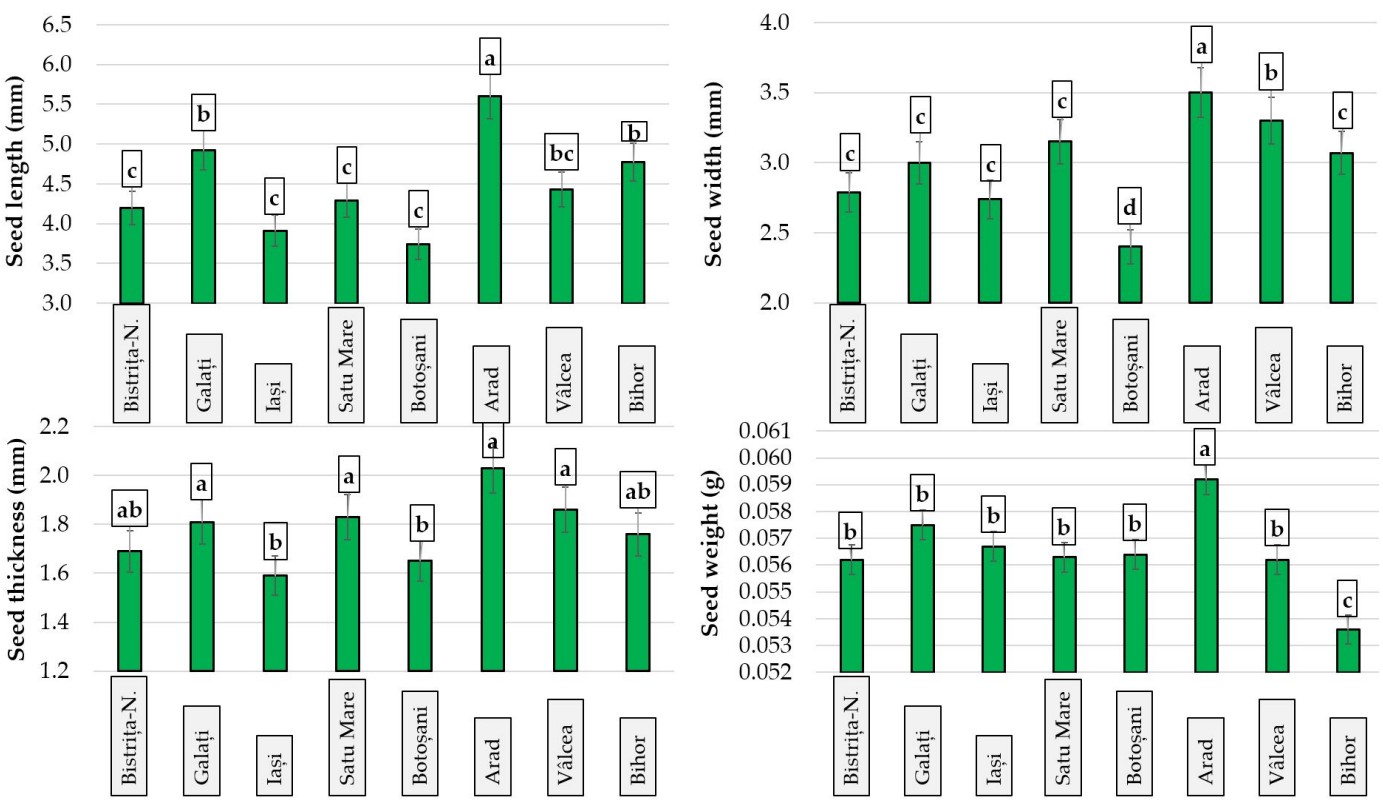

**Figure 3.** The principal traits of the seeds (as mean and SEM), based on the provenances of *R. pseudoacacia*. Different letters within means indicate statistically significant differences for the respective trait, at a significant level of *p* < 0.05 (Duncan test).

### 3.2. R. pseudoacacia Seed Germination

The evolution of *R. pseudoacacia* seed germination was analyzed for the studied provenances. For all treatments applied to stimulate germination, relatively ascending equations and respective regression lines are provided (Figure 4). The percentage of seed germination (%) during 15 days of investigation, depending on the germination stimulation treatment, per ensemble of the experience represented by the eight provenances, had an analogous evolution. In addition to the regression line and the regression equation, the coefficient of determination ($R^2$) and the Pearson correlation coefficient (r) are presented for each treatment, and the data were comparable. The scarification treatment was the one that showed the best germination results for the current study, followed by the thermal treatment at the highest temperature (100 °C) (GP = 41.7% and 33.9%, respectively). As was expected, the untreated seeds (control) recorded the lowest values for germination, with noticeably inferior data in comparison with all treatments.

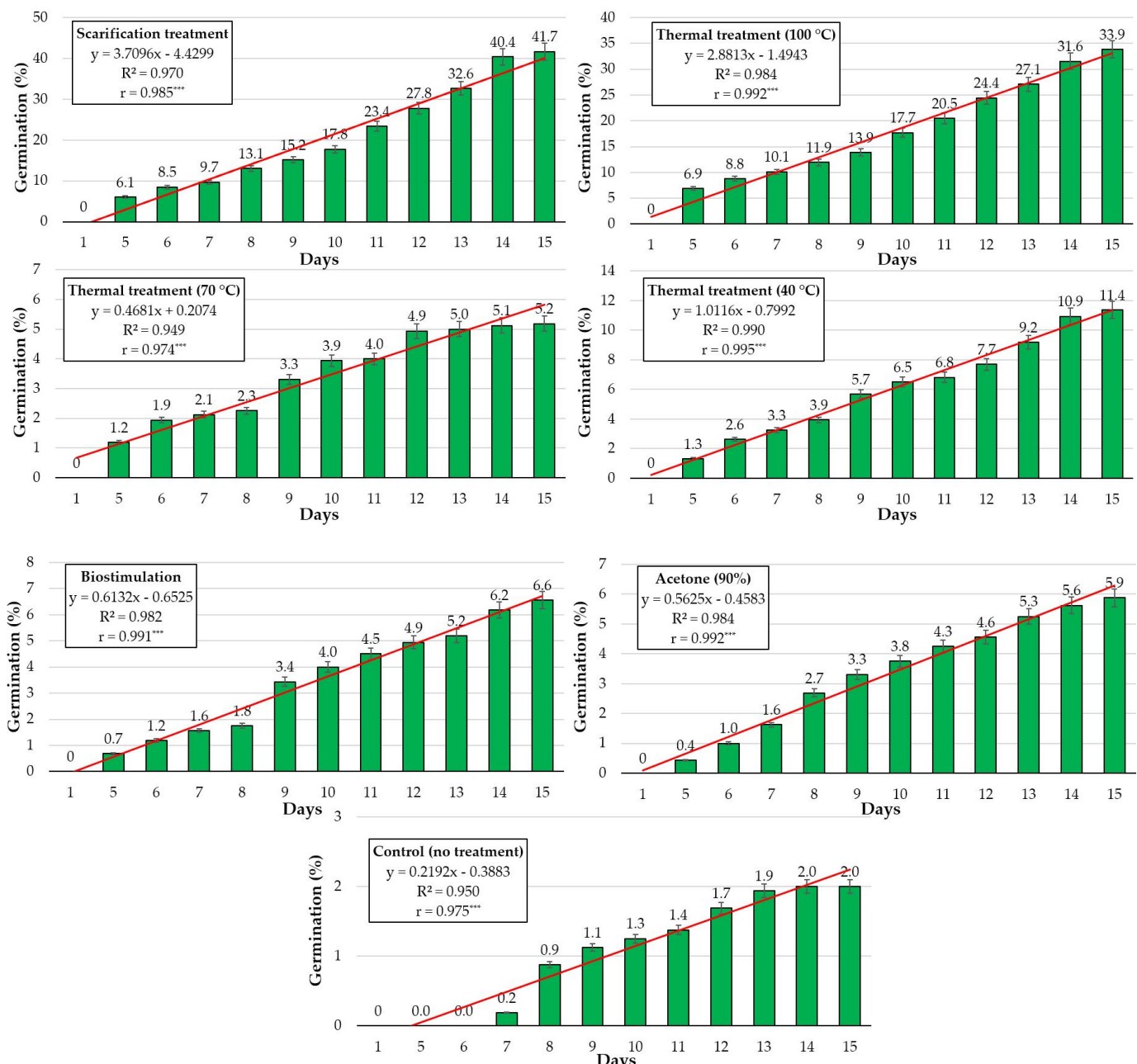

**Figure 4.** Seed germination progress (%) in days, depending on germination stimulation treatment, illustrated by regression line and regression equation, coefficient of determination ($R^2$), and Pearson correlation coefficient (r). The symbol *** after r-values means statistically significant, $p < 0.001$.

The regression line and equations between *R. pseudoacacia* germination and the main traits measured for the seeds as means of all provenances. Figure 5 shows how the independent variable, either seed length, width, thickness, or weight, influenced the "dependent" germination for $R^2 = 0.337$. In short, the regression shows how much germination can be expected to change as seed traits vary. The regression line was scaling, except in the case of seed thickness and germination, where the regression line was noted to be descending. The empirical data show that the established correlations among seed traits and germination were not significant.

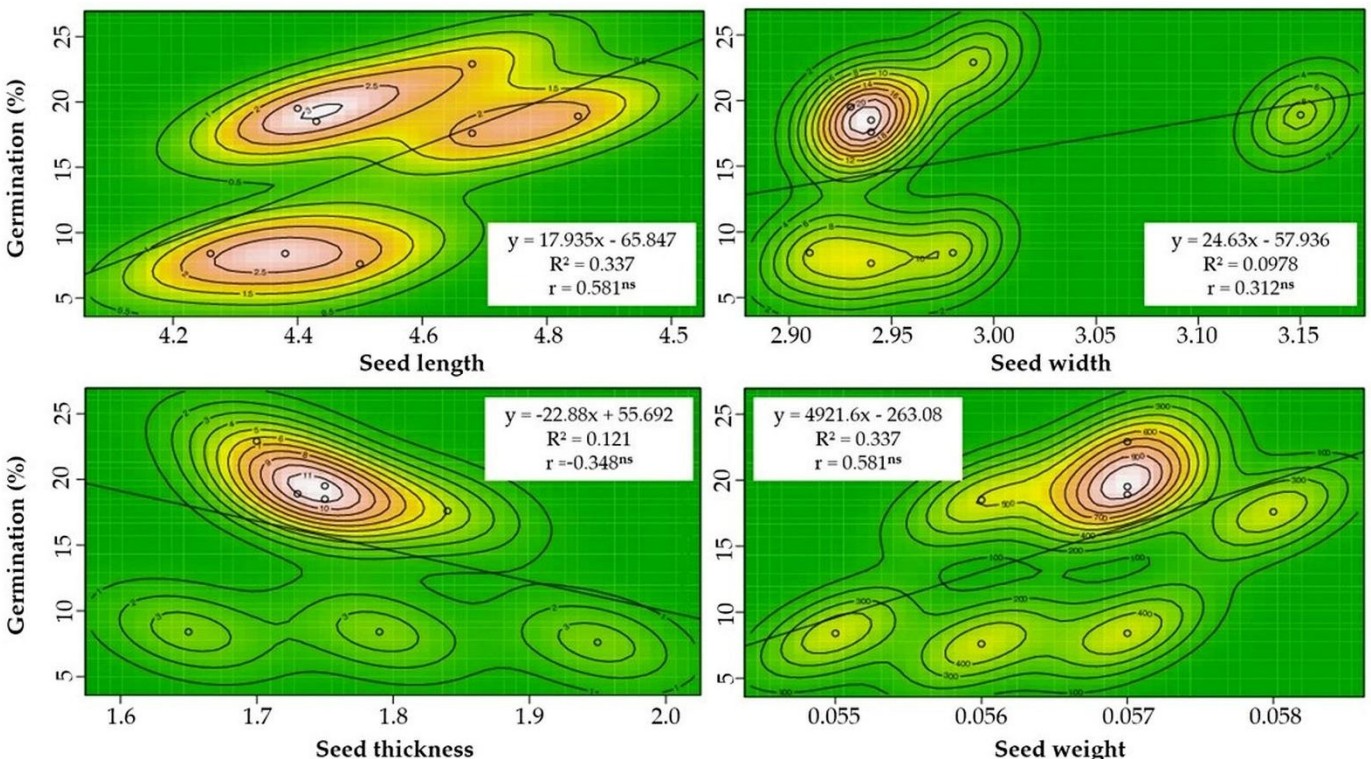

**Figure 5.** The regression line and equation between germination and the main traits of the seeds, the coefficient of determination ($R^2$), and the Pearson correlation coefficient (r) are presented for the entire experiment (as means of all provenances). Graphical representation was performed using bivariate kernel density estimation [31]; ns: not significant.

The use of germination data analysis based on specific indices is presented in Table 1. Germination percentage varied greatly, with values starting from 2% (Bihor provenance, with thermal treatment at 70 °C) up to 78.5% for the seeds from Iași provenance, after scarification. Overall, GP had higher values for the scarification treatment, followed by the thermal treatment at the highest temperature of 100 °C, whereas it was between 1.0% and 3.5% for control. It was noted that some provenances had distinct values within the same treatment, suggesting that the origin of the seeds can have a significant impact on germination (e.g., scarification data resulted in GP = 9% for Bihor provenance and had the highest value of 78.5%, followed by 68.5% for Iași and Galați provenances). GP and GI had the highest values for Iași provenance with scarification. The seedling vigor index (SVI), which takes into account the length of the seedlings and germination percentage, ranged from 0.3 and 4.71 with the scarification treatment.

It was surprisingly noted that the speed of emerge (SE) had high values for control, whereas it ranged from 2.9 (Satu-Mare provenance, scarification treatment) to 75.0 (Bihor provenance, thermal treatment at 70 °C) (Table 1). The coefficient of the rate of germination (CRG) and the seedling vigor index (SVI) did not have a similar trend with increasing the temperature for the thermal treatments within all analyzed *R. pseudoacacia* seed provenances; e.g., for Botoșani provenance, CRG = 12.8 with the thermal treatment at 100 °C, 11.5 at 70 °C, and 15.4 at 40 °C, and for Vâlcea provenance, CRG = 14.1 with the thermal treatment at 70 °C, 9.6 at 100 °C, 9.8 at 40 °C (Table 1). These variations may imply that the method chosen for germination stimulation must be strictly managed and the optimal protocol should be adapted.

Altogether, the mean for the whole experiment was calculated (Table 2) for each treatment to test whether one can decide upon the most appropriate method for germination stimulation. Scarification could significantly stimulate germination for *R. pseudoacacia*, and GP was the highest for this treatment and was followed by thermal treatment at 100 °C,

whereas thermal treatment at 70 °C, biostimulation, and acetone treatments did not differ among each other significantly in regard to GP. SE was significantly higher for thermal treatment at 70 °C, whereas CRG and SVI had the highest values for the scarification method once again, but different degrees of significance.

**Table 1.** The main germination indices * depending on the provenances of *R. pseudoacacia* and the seed germination stimulation treatments.

| No. | Treatment/Provenance | Germination Indices | | | | |
|---|---|---|---|---|---|---|
| | | GP | GI | SE | CRG | SVI |
| Control (no treatment) | | | | | | |
| 1 | Bistrița-N. | 1.5 | 0.3 | 33.3 | 10.3 | 0.05 |
| 2 | Galați | 1.0 | 0.2 | 50.0 | 11.8 | 0.03 |
| 3 | Iași | 2.0 | 0.5 | 75.0 | 10.8 | 0.08 |
| 4 | Satu-Mare | 2.5 | 0.4 | 20.0 | 8.2 | 0.08 |
| 5 | Botoșani | 3.0 | 0.7 | 66.7 | 10.5 | 0.11 |
| 6 | Arad | 2.5 | 0.5 | 60.0 | 10.8 | 0.08 |
| 7 | Vâlcea | 3.5 | 0.7 | 28.6 | 10.1 | 0.10 |
| 8 | Bihor | 1.5 | 0.2 | 33.3 | 8.1 | 0.06 |
| Scarification | | | | | | |
| 1 | Bistrița-N. | 55.0 | 14.5 | 32.7 | 11.2 | 3.03 |
| 2 | Galați | 68.5 | 12.6 | 4.4 | 8.6 | 1.40 |
| 3 | Iași | 78.5 | 22.1 | 25.5 | 12.6 | 4.71 |
| 4 | Satu-Mare | 51.0 | 8.9 | 2.9 | 8.3 | 2.55 |
| 5 | Botoșani | 10.5 | 2.5 | 14.3 | 10.7 | 0.74 |
| 6 | Arad | 11.0 | 2.2 | 9.1 | 9.0 | 0.61 |
| 7 | Vâlcea | 50.0 | 8.7 | 4.0 | 8.2 | 1.50 |
| 8 | Bihor | 9.0 | 2.1 | 16.7 | 10.1 | 0.30 |
| Thermal treatment (100 °C) | | | | | | |
| 1 | Bistrița-N. | 60.0 | 14.6 | 23.3 | 10.3 | 3.0 |
| 2 | Galați | 13.0 | 2.8 | 15.4 | 9.5 | 0.52 |
| 3 | Iași | 56.0 | 12.6 | 10.7 | 10.2 | 3.1 |
| 4 | Satu-Mare | 63.5 | 15.1 | 26.0 | 9.9 | 3.2 |
| 5 | Botoșani | 10.2 | 2.9 | 45.3 | 12.8 | 0.55 |
| 6 | Arad | 10.0 | 3.0 | 45.0 | 13.0 | 0.52 |
| 7 | Vâlcea | 41.5 | 9.2 | 18.1 | 9.6 | 1.33 |
| 8 | Bihor | 17.0 | 3.4 | 8.8 | 9.1 | 0.51 |
| Thermal treatment (70 °C) | | | | | | |
| 1 | Bistrița-N. | 5.0 | 1.3 | 20.0 | 11.9 | 0.23 |
| 2 | Galați | 6.0 | 1.5 | 8.3 | 11.2 | 0.18 |
| 3 | Iași | 6.0 | 1.6 | 33.3 | 11.8 | 0.33 |
| 4 | Satu-Mare | 6.3 | 1.5 | 33.5 | 11.9 | 0.25 |
| 5 | Botoșani | 6.1 | 1.6 | 8.4 | 11.5 | 0.19 |
| 6 | Arad | 6.2 | 1.7 | 7.9 | 11.4 | 0.30 |
| 7 | Vâlcea | 4.5 | 1.5 | 66.7 | 14.1 | 0.14 |
| 8 | 8 | 2.0 | 0.4 | 75.0 | 9.5 | 0.06 |
| Thermal treatment (40 °C) | | | | | | |
| 1 | Bistrița-N. | 2.5 | 0.6 | 40.0 | 11.1 | 0.11 |
| 2 | Galați | 33.5 | 6.2 | 6.0 | 8.6 | 1.07 |
| 3 | Iași | 7.5 | 1.7 | 13.3 | 9.7 | 0.34 |
| 4 | Satu-Mare | 5.0 | 1.3 | 20.0 | 11.6 | 0.25 |
| 5 | Botoșani | 8.1 | 2.8 | 25.2 | 15.4 | 0.32 |
| 6 | Arad | 8.0 | 2.6 | 25.0 | 15.2 | 0.40 |
| 7 | Vâlcea | 8.5 | 1.8 | 11.8 | 9.8 | 0.26 |
| 8 | Bihor | 18.0 | 4.1 | 8.3 | 10.3 | 0.72 |

**Table 1.** *Cont.*

| No. | Treatment/Provenance | Germination Indices | | | | |
|---|---|---|---|---|---|---|
| | | GP | GI | SE | CRG | SVI |
| Biostimulation | | | | | | |
| 1 | Bistrița-N. | 4.5 | 1.1 | 33.3 | 10.7 | 0.14 |
| 2 | Galați | 5.1 | 1.4 | 10.1 | 12.5 | 0.15 |
| 3 | Iași | 5.0 | 1.1 | 10.0 | 10.1 | 0.20 |
| 4 | Satu-Mare | 4.5 | 1.0 | 11.1 | 10.1 | 0.14 |
| 5 | Botoșani | 11.7 | 2.1 | 8.4 | 9.2 | 0.31 |
| 6 | Arad | 11.5 | 2.3 | 8.7 | 9.5 | 0.35 |
| 7 | Vâlcea | 7.5 | 2.1 | 40.0 | 11.9 | 0.23 |
| 8 | Bihor | 3.0 | 0.5 | 16.7 | 8.0 | 0.12 |
| Acetone (90%) | | | | | | |
| 1 | Bistrița-N. | 3.5 | 0.7 | 14.3 | 9.3 | 0.11 |
| 2 | Galați | 2.5 | 0.8 | 40.0 | 14.3 | 0.10 |
| 3 | Iași | 5.0 | 1.1 | 10.0 | 10.1 | 0.15 |
| 4 | Satu-Mare | 5.5 | 1.5 | 36.4 | 11.1 | 0.22 |
| 5 | Botoșani | 10.2 | 2.5 | 9.8 | 10.2 | 0.35 |
| 6 | Arad | 10.0 | 2.3 | 10.0 | 10.6 | 0.32 |
| 7 | Vâlcea | 7.5 | 1.7 | 6.7 | 10.6 | 0.23 |
| 8 | Bihor | 3.0 | 0.6 | 16.7 | 8.8 | 0.16 |

* GP—germination percentage, GI—germination index, SE—speed of emergence, CRG—coefficient of germination speed, SVI—seedling vigor index.

**Table 2.** The main germination index mean for the ensemble experience of eight provenances of *R. pseudoacacia* depending on the seed stimulation treatments.

| Treatment | Germination Indices | | | | |
|---|---|---|---|---|---|
| | GP | GI | SE | CRG | SVI |
| Control (no treatment) | 2.2 [e] | 0.4 [e] | 45.9 [a] | 10.1 [a] | 0.1 [e] |
| Scarification | 41.7 [a] | 9.2 [a] | 13.7 [e] | 9.8 [a] | 1.9 [a] |
| Thermal treatment (100 °C) | 33.9 [b] | 8.0 [b] | 24.1 [c] | 10.6 [a] | 1.6 [b] |
| Thermal treatment (70 °C) | 5.3 [d] | 1.4 [d] | 31.6 [b] | 11.7 [b] | 0.2 [d] |
| Thermal treatment (40 °C) | 11.4 [c] | 2.6 [c] | 18.7 [d] | 11.5 [b] | 0.4 [c] |
| Biostimulation | 6.6 [d] | 1.5 [d] | 17.3 [d] | 10.3 [a] | 0.2 [d] |
| Acetone (90%) | 5.9 [d] | 1.4 [d] | 18.0 [d] | 10.6 [a] | 0.2 [d] |

Means in columns for each index, followed by the same letter, are not significantly different at the $p < 0.05$ level according to Duncan's multiple range test (DMRT).

### 3.3. Seedling Survey and Growth Rate Depending on Provenances

In order to relate the origin of *R. pseudoacacia* seeds, the provenances, the studied germination treatments, and the growth in the first stages of plant development, the obtained seedlings were investigated.

The germination percentage of the seeds at the end of the 15 days of analysis showed that the scarification treatment provided the best results among all treatments (41.7%), with significant superior differences. Scarification was followed by the thermal treatment at 100 °C, even though the differences were significant between the two treatments (Figure 6). The same treatments were noted to promote the highest values also for seedlings length, but this time the differences were not significant, and what is more, were close for all results for the rest of the thermal treatments. Subsequently, acetone and biostimulation did not register statistically significant differences with the untreated seeds (control) for the seedling measurements. It will be interesting to further understand the different result obtained for the thermal treatment at 70 °C, where the germination was statistically inferior, but the seedlings recorded a superior value.

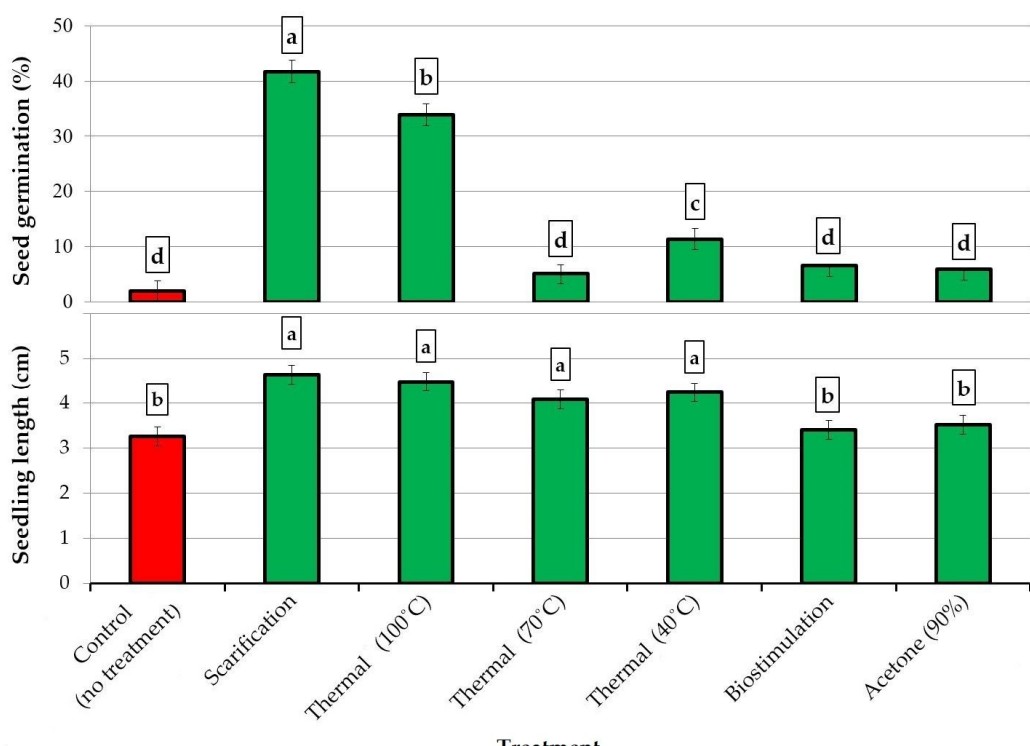

**Figure 6.** The germination percentage (mean ± SEM) and the length of the seedlings, depending on the *R. pseudoacacia* provenances. Means in columns for the provenances followed by the same letter were not significantly different at the $p < 0.05$ level according to Duncan's multiple range test (DMRT).

In order to have a more accurate representation of the *R. pseudoacacia* seedlings' growth and the possible influence of the seeds' origins, referring to the different provenances studied and secondly the germination treatments, the seedlings were consecutively measured in the first, second, and fourth year after planting. The data are interpreted in Tables 3 and 4. The main growth traits are expressed as means based on provenance, seedling height, and diameter. In the first year of life, seedlings originated from seeds obtained from Galați provenance were noted to have the highest height, 70.4 cm, which was statistically superior compared with the other provenances; what is more, CV% had the smallest value (20.2%) for Galați seedlings, and thus the trait was stable among the investigated plantlets (Table 3). As well for the diameter, Galați seedlings were superior, followed by those from Vâlcea and Arad. Furthermore, in the second year of life, Galați seedlings remained the highest ones, but were more closely followed by Iași, Vâlcea, Botoșani, and Arad ones, among which the differences were not significant. The highest values for diameter were noted among seedlings from Vâlcea and Galați. It is worth saying that in the second year of the study, CV% was smaller, ranging between 17.2% for height and 34% for diameter (Table 3); thus, the variability of the traits was small or medium and standard deviation was relatively small compared to the mean per provenance.

**Table 3.** Seedling height and diameter at the end of the first year and second year of life, depending on the geographical origin of the eight *R. pseudoacacia* provenances.

| No. | Provenance | Seedling Height (cm) | | Seedling Diameter (mm) | | Seedling Height (cm) | | Seedling Diameter (mm) | |
|---|---|---|---|---|---|---|---|---|---|
| | | First Year | | First Year | | Second Year | | Second Year | |
| | | Mean * ± SEM | CV% | Mean ± SEM | CV% | Mean ± SEM | CV% | Mean ± SEM | CV% |
| 1 | Bistrița-N. | 24.6 [c] ± 0.9 | 39.4 | 1.4 [c] ± 0.1 | 42.0 | 46.2 [c] ± 1.0 | 21.2 | 5.7 [c] ± 0.1 | 20.7 |
| 2 | Galați | 70.4 [a] ± 1.4 | 20.2 | 5.0 [a] ± 0.2 | 44.5 | 87.6 [a] ± 1.8 | 20.1 | 10.8 [a] ± 0.4 | 34.0 |
| 3 | Iași | 53.7 [b] ± 2.1 | 39.7 | 3.6 [ab] ± 0.2 | 52.2 | 82.3 [a] ± 1.4 | 17.2 | 7.8 [b] ± 0.2 | 23.1 |
| 4 | Satu Mare | 29.6 [c] ± 1.5 | 51.3 | 2.3 [bc] ± 0.1 | 43.2 | 57.4 [b] ± 1.3 | 22.7 | 7.4 [b] ± 0.2 | 28.2 |
| 5 | Botoșani | 46.0 [b] ± 1.8 | 39.6 | 3.2 [b] ± 0.1 | 42.9 | 77.4 [a] ± 1.3 | 17.2 | 8.2 [b] ± 0.2 | 20.8 |
| 6 | Arad | 45.2 [b] ± 1.6 | 34.9 | 2.8 [b] ± 0.1 | 47.1 | 76.9 [a] ± 1.3 | 16.8 | 7.5 [b] ± 0.2 | 23.3 |
| 7 | Vâlcea | 50.7 [b] ± 1.7 | 34.2 | 4.4 [a] ± 0.2 | 45.8 | 77.4 [a] ± 1.4 | 18.4 | 10.9 [a] ± 0.3 | 26.2 |
| 8 | Bihor | 41.8 [b] ± 1.9 | 45.2 | 1.8 [c] ± 0.1 | 59.7 | 62.2 [b] ± 1.3 | 21.7 | 6.7 [bc] ± 0.2 | 25.8 |

\* Means in columns for the provenances, followed by the same letter, are not significantly different at the $p < 0.05$ level according to Duncan's multiple range test (DMRT).

**Table 4.** Seedling height, diameter, and the number of plant ramifications at the end of the fourth year of life, depending on the geographical origin of the eight *R. pseudoacacia* provenances.

| No. | Provenance | Seedling Height (cm) | | Seedling Diameter (mm) | | Number of Ramifications per Plant | |
|---|---|---|---|---|---|---|---|
| | | Mean * ± SEM | CV% | Mean ± SEM | CV% | Mean ± SEM | CV% |
| 1 | Bistrița-N. | 65.0 [d] ± 1.7 | 26.2 | 6.8 [d] ± 0.2 | 24.8 | 6.1 [b] ± 0.1 | 19.8 |
| 2 | Galați | 274.0 [a] ± 4.8 | 17.5 | 31.6 [a] ± 0.5 | 16.3 | 10.1 [a] ± 0.2 | 19.0 |
| 3 | Iași | 203.5 [b] ± 6.6 | 32.6 | 15.7 [c] ± 0.6 | 41.0 | 7.4 [b] ± 0.2 | 21.7 |
| 4 | Satu Mare | 213.5 [b] ± 3.7 | 17.5 | 14.8 [c] ± 0.6 | 38.8 | 6.8 [b] ± 0.1 | 18.8 |
| 5 | Botoșani | 258.0 [a] ± 5.4 | 20.9 | 29.9 [a] ± 0.9 | 29.6 | 9.7 [a] ± 0.2 | 20.5 |
| 6 | Arad | 116.5 [c] ± 3.4 | 29.3 | 13.0 [c] ± 0.4 | 30.3 | 7.2 [b] ± 0.1 | 19.9 |
| 7 | Vâlcea | 267.0 [a] ± 5.1 | 18.9 | 23.3 [b] ± 0.5 | 23.2 | 9.4 [a] ± 0.2 | 18.4 |
| 8 | Bihor | 278.0 [a] ± 6.6 | 23.7 | 25.3 [b] ± 0.6 | 25.1 | 6.3 [b] ± 0.1 | 13.7 |

\* Means in columns for the provenances, followed by the same letter, are not significantly different at the $p < 0.05$ level according to Duncan's multiple range test (DMRT).

The fourth year of the investigation favored the calculation of one more trait that was influential for the seedlings' development: the number of ramifications (Table 4). The black locust seedlings' height varied from 65 cm (Bistrița-Năsăud) to 278 cm (Bihor). Galați seedlings had statistically assured superior values for height (the second mean), diameter (highest mean and followed only by Botoșani seedlings, with no significant differences), and number of ramifications (highest mean, followed by Botoșani and Vâlcea). Growth parameters for Galați seedlings were constant, as the CV% was always small (less than 20) in the calculations provided.

*3.4. The Overall Influence of the Provenances*

It is impressive to understand the impact that the origin from different provenances and the germination stimulation can have upon the development of seedlings, as *Robinia* is a genus of forestry interest, and thus reproductive material of high quality is essential for proper management.

The relationship between seed origin and parameters, germination, and seedling traits is highlighted by the Pearson correlation (Figure 7). Positive correlations, which showed a directional dependence, are displayed in blue and negative correlations in red. The color intensity and the size of the circle are proportional to the correlation coefficients. The grey background boxes present in some cells illustrate the statistically assured values ($p < 0.05$). Deriving from the association between variables, the investigated traits of thickness and

width of the seeds were positively correlated and statistically assured, but not with the germination data. What is more, seedlings length and diameter, especially in the second year of life, were also statistically positively assured.

| Correlation among analysed traits | Seed length | Seed width | Seed thickness | Seed weight | Germination % | Seedlings length G% | Seedlings length 1y | Seedlings length 2y | Seedlings length 4y | Seedlings diameter 1y | Seedlings diameter 2y | Seedlings diameter 4y | No ramifications 4y |
|---|---|---|---|---|---|---|---|---|---|---|---|---|---|
| Seed length | 1.00 | 0.82 | 0.87 | -0.19 | -0.38 | -0.40 | 0.22 | 0.16 | -0.16 | 0.06 | 0.12 | -0.04 | -0.11 |
| Seed width | 0.82 | 1.00 | 0.90 | -0.09 | -0.09 | -0.50 | 0.02 | 0.02 | -0.11 | 0.07 | 0.20 | -0.24 | -0.19 |
| Seed thickness | 0.87 | 0.90 | 1.00 | 0.05 | -0.38 | -0.35 | 0.04 | 0.09 | -0.13 | 0.10 | 0.23 | -0.09 | 0.00 |
| Seed weight | -0.19 | -0.09 | 0.05 | 1.00 |  | 0.27 | 0.10 | 0.25 | -0.35 |  | 0.30 | -0.24 |  |
| Germination % | -0.38 | -0.09 | -0.38 |  | 1.00 | -0.13 | 0.07 | -0.01 | -0.13 | 0.28 | 0.20 | -0.32 | 0.01 |
| Seedlings length G% | -0.40 | -0.50 | -0.35 | 0.27 | -0.13 | 1.00 | -0.42 | -0.16 | -0.42 | -0.43 | 0.61 | -0.36 | -0.29 |
| Seedlings length 1y | 0.22 | 0.02 | 0.04 | 0.10 | 0.07 | -0.42 | 1.00 | 0.93 |  | 0.90 | 0.80 | 0.70 | 0.75 |
| Seedlings length 2y | 0.16 | 0.02 | 0.09 | 0.25 | -0.01 | -0.16 | 0.93 | 1.00 |  | 0.89 | 0.76 |  | 0.76 |
| Seedlings length 4y | -0.16 | -0.11 | -0.13 | -0.35 | -0.13 | -0.42 |  |  | 1.00 |  |  | 0.89 |  |
| Seedlings diameter 1y | 0.06 | 0.07 | 0.10 |  | 0.26 | -0.43 | 0.90 | 0.89 |  | 1.00 | 0.95 |  | 0.88 |
| Seedlings diameter 2y | 0.12 | 0.20 | 0.23 | 0.30 | 0.20 | 0.61 | 0.80 | 0.76 |  | 0.95 | 1.00 |  | 0.88 |
| Seedlings diameter 4y | -0.04 | -0.24 | -0.09 | -0.24 | -0.32 | -0.36 | 0.70 |  | 0.89 |  |  | 1.00 | 0.77 |
| No ramifications 4y | -0.11 | -0.19 | 0.00 |  | 0.01 | -0.29 | 0.75 | 0.76 |  | 0.88 | 0.88 | 0.77 | 1.00 |

**Figure 7.** Pearson correlations (r-value) among the analyzed characteristics of the eight *R. pseudoacacia* provenances. The correlation was significant at the 0.05 level (2-tailed), and in the boxes, the assured correlations are marked with a grey background.

The strength of the relationship for the traits investigated was evidenced by positively statistically assured data among seedling length and diameter in the first two years of growth, and with the number of ramifications in the fourth year. Values closer to r = 1, meaning a stronger relationship between the variables, was seen among vegetative growth, length, and diameter. The positive correlation coefficient indicates that an increase in one variable would correspond to an increase within the associated one, thus implying a direct relationship between those two variables (e.g., as seen for seedling length and diameter), whereas a negative correlation indicates an inverse dependency, meaning one variable increases and the second one decreases [33].

Canonical correspondence analysis (CCA) is a multivariate method useful for elucidating and representing the relationships between biological assemblages of species and their environment [34]. In this study, traits investigated for seeds included germination treatments, seedling characteristics, and provenance of origin. Canonical correspondence analysis (CCA), illustrated in Figure 8, provided a presentation for the eight *R. pseudoacacia* provenances and 13 analyzed characteristics of the seeds and the seedlings obtained. By identifying new variables as linear functions of those in the original data set, it was found that the first axis of the CCA, Axis 1, accounted for 73.65%, whereas the second component (Axis 2) accounted for 20.51% of the total variation observed. The most distant provenance was Bistrița-Năsăud, which was in negative correlation with Bihor.

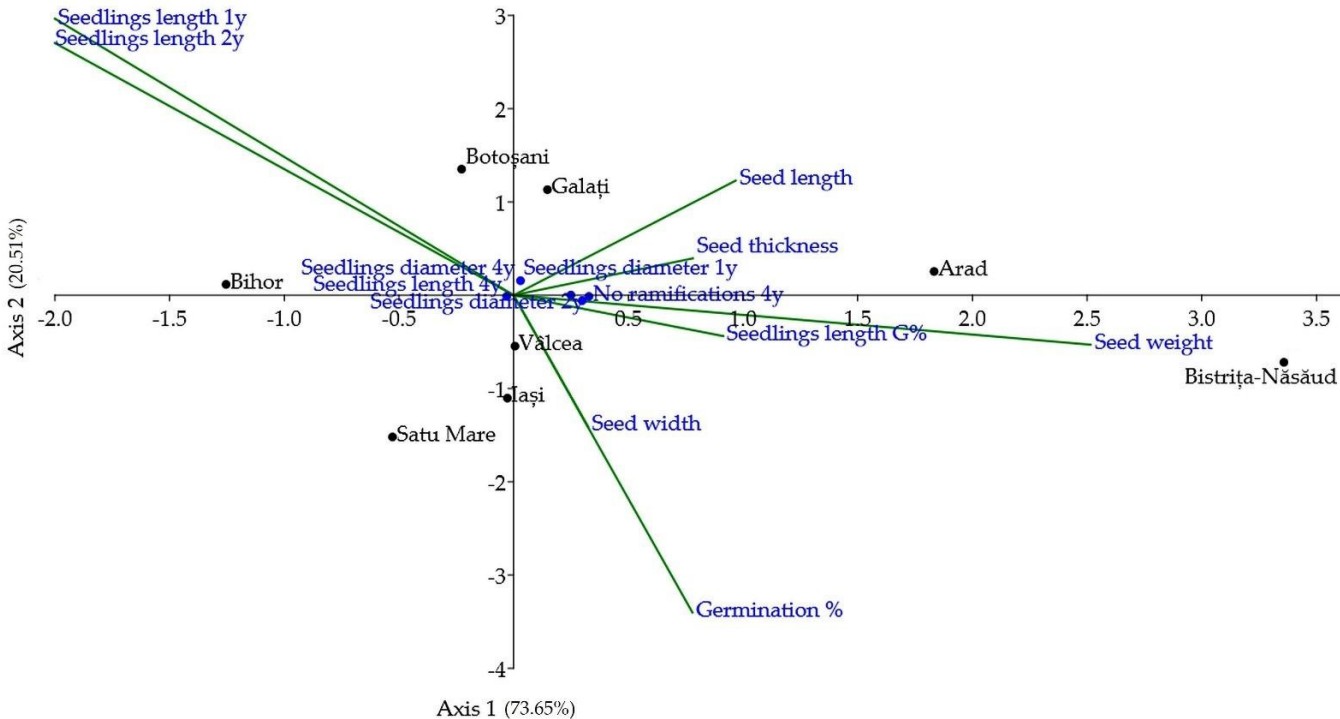

**Figure 8.** Canonical correspondence analysis (CCA) for the eight *R. pseudoacacia* provenances and 13 analyzed characteristics of the seeds and seedlings obtained.

Data were ordinarily dispersed within the investigated traits and provenances (Figures 8 and 9). In the case of the analyzed features, a distinct and obvious cluster was observed for seedling length in the fourth year of investigation. Another cluster consisted of seedling length in the first and second year of life, separated by a third cluster, including many other distinct subclusters.

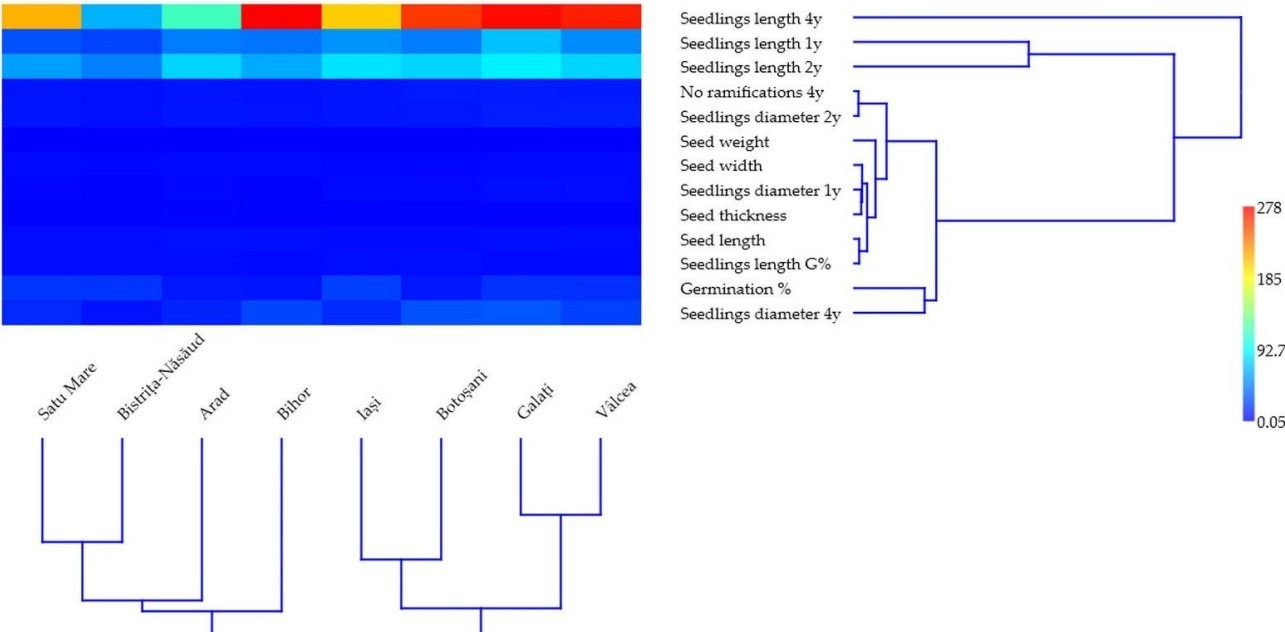

**Figure 9.** Hierarchical clustering—paired group UPGMA (unweighted pair group method with arithmetic mean)—similarity index (Gower) of the eight *R. pseudoacacia* provenances and 13 analyzed characteristics of the seeds and seedlings obtained.

Furthermore, the obtained dendrogram (Figure 9) shows the clusters of the provenances according to the grouping of the observations made and the levels of similarity among all traits investigated hereby. The tree diagram confirms the dispersed results, both by the pattern by which the clusters were formed and by the similarity or distance levels of the clusters that contained the analyzed *R. pseudoacacia* provenances. There were two large clusters, both with several subclusters. It was noted that Iași and Botoșani were grouped together in one subcluster, probably due to the data about germination, whereas Bihor stood alone as a separate subcluster in one part of the tree, as the seedlings originated from seeds obtained from its seeds registered significantly superior results.

## 4. Discussion

Morphological traits noted for black locust seeds indicated that the origin was relevant for the current investigation, with the data being quite variable [6]. The present study revealed seed traits such as thickness, length, width, and weight, which are represented as a boxplot diagram (Figure 2) consisting of the minimum and maximum range values, the upper and lower quartiles, as well as the median, to summarize the distribution of the dataset recorded. It is important to mention that the seeds from Bihor provenance had a relatively low variability for all analyzed traits. For Vâlcea and Galați provenances, the mean of the empirical data was close to the median in the box plot for seed length and thickness. The differences among provenances analyzed were statistically assured, so the germination was surely influenced by the ecological parameters in the area of provenance, along with the applied treatment.

According to the literature, higher germination may be identified in larger seeds because they contain a greater reserve of nutritional and energetic resources that support the germination [35] and subsequently healthier seedlings [25,36]. Black locust seeds are characterized as hard seeds that are impermeable to water. Thus, for this species, the natural dormancy of the seeds is entirely physical (seed hardness) [9,11]. It is characteristic of several plant families, one of the largest of which is Fabaceae [18]. To break the physical rest of the seeds, the activity of microorganisms, high temperature (fires), and low temperature (frost) are needed [37,38]. When the seed tegument is damaged, water infiltrates and initiates the germination process. For the *Robinia* genus, the proportion of seeds capable of germination without any treatment is usually low, as evidenced by the average level of 10.8% obtained in the past [39]. The obtained results also showed a very low germination (2.18%) without any treatment. Similar results were obtained by several other researchers [40,41].

For economic purposes, the scarification treatment is applied to the seeds before sowing to increase the germination rate. Common methods include treating the seeds in water baths at temperatures ranging from 60 to 98 °C [40], with sulfuric acid [42], or with methods that physically damage the seed coat [41,43]. There are various chemical and mechanical treatments also used to stimulate germination. The germination capacity of black locust seeds is facilitated by submersion in boiling water, mechanical treatments (scarification), or by chemical treatments (e.g., using diluted sulfuric acid) before sowing [44,45].

The evolution of seed germination (%) during the experiment, depending on the germination stimulation treatment per ensemble of the study, represented by the eight provenances for the origin of *R. pseudoacacia* seeds, was similar. In addition to the regression line and the regression equation, the coefficient of determination ($R^2$) and the Pearson correlation coefficient (r) were presented for each treatment. The scarification treatment showed the best germination results for the current analysis.

Comparable results were obtained by other authors [9], with high germination rates obtained by immersing the seeds in boiled water at temperatures of 60–80 °C with a soaking time of 20 min to 72 h. In the current study, GP had higher values for the scarification treatment, followed by the thermal treatment at the highest temperature of 100 °C and a soaking time of 10 min. In forestry practice, it may also be sufficient to use a chemical method by immersing the seeds for 60 min in concentrated sulfuric acid, then washing

the seeds in cold water and leaving them to dry at adjusted temperature [46]. Another successful method is by scarifying the seeds using manual abrasion sandpaper so that the seed coat is scratched [15,17,47].

Hence, the germination investigation in this study also confirmed the best germination capacity of black locust seeds after scarification treatment, similar to recently reported results within the specialized literature, with GR up to and above 95% [9,17,48]. Even so, these authors found that there were no significant differences in GR between scarification treatments, with preliminary tests suggesting that mechanical scarification provided the highest germination rates, as observed in many other Fabaceae species [18]. In our study, the data obtained for the germination was positively statistically assured, with higher values for GP and GI, as well as SVI (Table 2) for scarification compared with thermal treatments, biostimulation, or acetone solution.

CVG does not focus on the final percentage of germination, but places emphasis on the time required to reach it. The details of time (first day, last day, and time spread) are not taken into account, as the time is averaged. One study [16] showed seed lots with the same FDG, LDG, and TSG but different CVG values. This means that time-based measurements not correlated with supplementary data are not a very useful representation of the overall seed germination activity. Starting germination and ending it at the same time is not sufficient to produce a uniform CVG, and therefore it can be misleading.

The GI appears to be the most comprehensive measurement parameter, combining both germination percentage and speed (spread, duration, and high/low events). It magnifies the variation among seed lots in this regard with an easily compared numerical measurement. The coefficient of the rate of germination (CRG) and the seedling vigor index (SVI) significantly decreased with increasing concentrations of extracts in relation to control values, irrespective of the time of treatment of seeds and type of extract [24].

The use of germination data analysis based on indices showed that the germination percentage varied greatly, with values from 2% (Bihor provenance, with thermal treatment at 70 °C) up to 78.5% for the seeds from Iași provenance after scarification. Overall, GP had higher values for the scarification treatment, followed by the thermal treatment at the highest temperature of 100 °C, whereas it was between 1.0% and 3.5% for control. It was noted that some provenances had distinct values within the same treatment, suggesting that the origin of seeds can have a significant impact on germination (e.g., scarification data resulted in GP = 9% for Bihor provenance and had the highest value of 78.5%, followed by 68.5% for Iași and Galați provenances). GP and GI had the highest values for Iași provenance with scarification.

It will be interesting to further understand the different results obtained for thermal treatment at 70 °C, where the germination was statistically inferior but the seedlings recorded superior values, similar to scarification, for example (Figure 5). Such a difference may be due to the adaptability of seeds, further directed to seedling development, taking into account the provenances from various climatic conditions, plus trees of diverse ages.

The height of the seedlings in the last year of investigation was statistically inferior in Bistrița-Năsăud provenance. The small height compared with the other provenances may have been due either to consanguinity among the seedlings, or could have been influenced by the origin of the seeds, which may have been collected from other trees that were not correctly selected or were genetically debilitated [49–51].

Canonical correspondence analysis (CCA) is a method for arranging species along environmental variables. CCA constructs linear combinations of environmental variables along which the distributions of the species are maximally separated. The eigenvalues produced by CCA measure the separation [34], pointing out the distributions of species along the environmental variables. Ecological applications demonstrate that CCA can be used both for detecting species–environment relations and for investigating specific questions about the response of species to environmental variables based on spatial processes that create autocorrelation, where the spatial structure of environmental factors creates spatial dependence [52]. Even more importantly, ordination methods like CCA are needed to

incorporate scale-dependent species–environment correlations. In our study, the canonical correspondence analysis (CCA) illustrated in Figure 8 provides a presentation for the eight *R. pseudoacacia* provenances and 13 analyzed characteristics of the seeds and seedlings obtained. Furthermore, the tree diagram (Figure 9) confirms the dispersed results, both by the pattern by which the clusters were formed and by the similarity or distance levels of the clusters that contain the analyzed *R. pseudoacacia* provenances.

## 5. Conclusions

The present results suggest that treatments applied to *R. pseudoacacia* seeds might improve their germination capacity and the seedlings' features, thus leading to more valuable reproductive biological material of the species. Stimulation of germination through physical methods (scarification) might be an alternative approach to the chemical substances used nowadays, with ecological advantages as well as the possibility to be used on a larger scale with high efficiency. The germination tests performed in the current study and the analysis of the indices revealed a remarkable variability in the germinal responses of seeds from different Romanian provenances. The geographical origin of the provenances from which the seeds belong was shown to significantly influence the development of future plants in terms of their growth rate in the very young stages of life. Differences among provenances for certain traits of reproductive material may also be due to some extent to seed providers in reservations included in the national catalogue, which probably do not always provide biological material at high-level standards. Consequently, similar studies can also be used to verify the quality of the biological material provided. The close correlations between some characteristics could be used as indirect selection indices and used effectively for the selection of valuable specimens in the nursery, to be used for afforestation or other purposes. The identified provenances of quality reproductive material could be recommended for use in new afforestation programs of black locust or for new breeding and selection work.

**Author Contributions:** Data curation, A.M.R.; formal analysis, A.M.T., C.D., R.E.S. and L.H.; investigation, A.M.T. and I.M.M.; methodology, A.M.T., I.M.M. and C.D.; project administration, R.E.S.; resources, A.F.S.; software, A.F.S.; supervision, M.B. and R.E.S.; validation, L.H. and R.E.S.; visualization, I.M.M., A.M.T. and O.V.; writing—original draft, C.D. and O.V.; writing—review and editing, C.D., I.M.M. and A.F.S. All authors have read and agreed to the published version of the manuscript.

**Funding:** This research was funded by University of Agricultural Sciences and Veterinary Medicine Cluj-Napoca (UASVM), grant number 26011/16.12.2020. The research was partly sustained by the Doctoral School from the UASVM during the Ph.D. study stage granted to A.M.R.

**Institutional Review Board Statement:** Not applicable.

**Informed Consent Statement:** Not applicable.

**Conflicts of Interest:** The authors declare no conflict of interest.

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
