# Peer review of "From Seed to Seedling: Influence of Seed Geographic Provenance and Germination Treatments on Reproductive Material Represented by Seedlings of Robinia pseudoacacia"

_sustainability, doi:10.3390/su14095654_

Round 1
Reviewer 1 Report
Title:
For me ‘reproductive material characteristics’ in the title was somewhat unclear and awkward. Could the authors say something simpler like ‘…influence of seed geographic provenance, traits, and germination treatments of Robinia pseudoacacia’?
Abstract:
The three factors studied (seed geographic provenance, seed traits, germination treatments) should be organized in the same order throughout the manuscript. As an example, the title lists an order that is not the same as the order in the first sentence of the abstract.
A more concise start would be ‘We investigated…’
The justification for the research (its importance) should be included in the abstract.
The sentence ‘Scarification was followed by thermal treatment at 100 °C, even though the differences were significant between the two treatments’ is unclear. I think what the authors mean to say is that seeds soaked in 100 °C water experienced the second-greatest germination rates.’
Introduction
Be more specific about the native range of black locust. It’s native range is not throughout North American. It has been introduced throughout North America.
Similar to the abstract, the importance/justification for the work is not well developed in the Introduction. The species is simultaneously presented as being invasive (i.e., a generally negative property) but useful for controlling soil erosion, etc. (i.e., postive properties). The rationale for the study is thus confusing and needs to be made clearer – especially within the realm of Sustainability.
The Introduction ends by framing the present study as one investigating ‘data analysis methods’ but the study described is experimental research. As such, the study needs a better description at the end of the Introduction for a more logical transition into the Methods and Results.
Were they any a priori hypotheses about the seed geographic provenance, traits, and germination treatments that guided the research? Or was it purely exploratory? If the former, then these hypotheses should be described in the Introduction. If the latter, then justification for the exploratory nature of the work should be provided.
Methods
More details about seed collection are needed. How many seeds were obtained from each tree and how many trees per provenance were sampled? Were parents tracked throughout the experiment or were seeds binned (combined) for each provenance? Were all seeds sampled from similar types of habitats? Without knowing how many trees/locations were sampled in each provenance, it is difficult to interpret the provenance-level differences in the study.
In the sentence that ends right above Figure 1, ‘ramifications’ needs to be defined.
It is not clear what the ellipsis (…) means in the equation for Germination Index.
Was differences in germination also investigated (as well as seed traits)? Were provenance, seed traits, and germination treatments treated as independent variables in the statistical analysis and germination as the dependent variable? This isn’t clear.
It looks like linear regression results are presented (Figure 4) but this analysis is not described in the Methods.
Results:
The first sentence of the results reads like a figure legend. The figure should be cited and the results (but not the figure) described in the text.
Are results presented in the text (such as in the first paragraph) means plus/minus standard errors? If so the SEs should be given.
Were the geographic locations and/or topography of provenances considered in the analysis (i.e., N-S, E-W, elevation differences, etc.)? What is the order in which the provenances are listed (from left to right) on the x-axes in Figure 2?
The use of the word ‘evolution’ in lines 221-224 is unclear.
What are the x-axes values in Figure 4 (1-15)? Are these days? The figure legend needs to be more descriptive. Are the r-values presented here p-values? Was the gemination success presented here additive? If so, wouldn’t these very significant regressions be expected (i.e., more germination over time)? If this is the case, it would be odd to test these results statistically.
I suggest that the information in tables be presented in a Supplment and then these results be summarized in the main body of the paper. In general, the Results section should be significantly shortened by presenting extraneous results in a supplement. The heavy results section results in a loss of focus of the study and makes the research seem overly exploratory.
Discussion:
The Discussion starts with a reiteration of results rather than with a discussion of the results. And this is a circumstance that is continued throughout the Discussion section. The authors should avoid repeating results in the Discussion and focus on interpreting the results within the context of the broader topic (i.e., questions asked, hypotheses addressed, application, future recommendations, etc.).
A citation is needed to support the assertion that the dormancy of black locust seeds is entirely physical as this was not tested in the study (lines 427-428).
Given the information presented in lines 437-444 and 451-459, the novel nature of the present study is unclear. Results should be discussed such that it is clear how they expand what is already known about the species. What is primarily discussed is that the results of the present study confirm previous results. In addition, the novel nature of the present student should be described in the Introduction section.
Author Response
Title
Q- For me ‘reproductive material characteristics’ in the title was somewhat unclear and awkward. Could the authors say something simpler like ‘…influence of seed geographic provenance, traits, and germination treatments of Robinia pseudoacacia’?
R- Thank you very much for the pertinent and exhaustive review. Your recommendations have been extremely helpful to us in improving the manuscript. Indeed, the improvement of the title is welcome, thank you. Based on your suggestions, we think it's much clearer and more comprehensive now. In addition, we tried to maintain in the title a touch of attractiveness for the readers of the journal ("From seed to seedling: ...").
Abstract
Q- The three factors studied (seed geographic provenance, seed traits, germination treatments) should be organized in the same order throughout the manuscript. As an example, the title lists an order that is not the same as the order in the first sentence of the abstract.
A more concise start would be ‘We investigated…’
The justification for the research (its importance) should be included in the abstract.
The sentence ‘Scarification was followed by thermal treatment at 100 °C, even though the differences were significant between the two treatments’ is unclear. I think what the authors mean to say is that seeds soaked in 100 °C water experienced the second-greatest germination rates.’
R- We rephrased as indicated. The order in the title and in the manuscript follows the chronological and phenotypical process. We intended to present the vegetative phases in a concise manner, so that we can respect the limited characters accepted in the abstract. We added instead the importance of the investigation, and we appreciate the remark.
Introduction
Q-Be more specific about the native range of black locust. It’s native range is not throughout North American. It has been introduced throughout North America.
The species is simultaneously presented as being invasive (i.e., a generally negative property) but useful for controlling soil erosion, etc. (i.e., postive properties). The rationale for the study is thus confusing and needs to be made clearer – especially within the realm of Sustainability.
R- The species originates from East of N.A. and was further introduced in Europe, with several benefits, but in the same time the invasiveness of the species was noted. In this regard, we mentioned the most important utilities, but also that the species management should consider its spread and the negative connotations for some. Robinia is known to have valuable uses, but is also considered invasive, despite its positive properties. Exactly because of the current trend, we mentioned “in the perspective of planning its management and biodiversity regulation, the frequent use nowadays and its invasive spread, all stages starting with seed harvest, to germination and further trees’ growth, must be considered”. “Positive economic but negative environmental impacts of Robinia result in conflicts of interest between nature conservation, forestry, urban landscaping, beekeepers and the public when defining management priorities. Because current legislation will determine the future distribution of Robinia in the landscape, a comprehensive view of this species is necessary.“
Even more, the authors intended to focus more on the seeds dormancy and germination within introduction, as the research on the species is not at the beginning, and thus other manuscripts, published before, detail the origin, utilities, risks associated with Robinia.
Similar to the abstract, the importance/justification for the work is not well developed in the Introduction. The Introduction ends by framing the present study as one investigating ‘data analysis methods’ but the study described is experimental research. As such, the study needs a better description at the end of the Introduction for a more logical transition into the Methods and Results.
Were they any a priori hypotheses about the seed geographic provenance, traits, and germination treatments that guided the research? Or was it purely exploratory? If the former, then these hypotheses should be described in the Introduction. If the latter, then justification for the exploratory nature of the work should be provided.
R- Modifications and additions were made in order to clear these aspects. The article has data analysis, using the results obtained from different treatments, which represent an experimental research, so that is a combination of data analysis and empiric research.
Methods
More details about seed collection are needed. How many seeds were obtained from each tree and how many trees per provenance were sampled? Were parents tracked throughout the experiment or were seeds binned (combined) for each provenance? Were all seeds sampled from similar types of habitats? Without knowing how many trees/locations were sampled in each provenance, it is difficult to interpret the provenance-level differences in the study.
R- We rephrased accordingly, as the authors did not directly collect the seeds. The seeds were collected as previously mentioned, from different providers from Romanian seed source reservations, corresponding to the Romanian Gene Reserved Forests and Seed Stands included in National Catalogue of Forest Genetic Resources and Forest Reproductive Materials. We procured the specific quantity of seeds, collected from each population from mature trees, but we do not have more details. The influence of the provenance (seeds origin) is considered in regard with the ecological differences.
In the sentence that ends right above Figure 1, ‘ramifications’ needs to be defined. R- Adequate explanation was added, thank you for the notification.
It is not clear what the ellipsis (…) means in the equation for Germination Index.
R - Determinations of germination index was made for 15 days, so we used (…) to summarise the formula. We used the formula and the explanations accepted and used in other published articles.
Was differences in germination also investigated (as well as seed traits)? Were provenance, seed traits, and germination treatments treated as independent variables in the statistical analysis and germination as the dependent variable? This isn’t clear.
R- Thank you. We have added the necessary information so that all the analyzes performed and the figures presented are clear (based on your recommendations, we have replaced four figures, all to be self-explanatory). In addition, it is now obvious which are the independent and respectively dependent variables in the regressions performed.
It looks like linear regression results are presented (Figure 4) but this analysis is not described in the Methods.
R – Regression was added to methodology, determined for germination and number of days.
Results
The first sentence of the results reads like a figure legend. The figure should be cited and the results (but not the figure) described in the text.
Are results presented in the text (such as in the first paragraph) means plus/minus standard errors? If so the SEs should be given.
R- Thank you. We solved and we have added all the necessary information (as data, we used xÌ… ± SEM)
Were the geographic locations and/or topography of provenances considered in the analysis (i.e., N-S, E-W, elevation differences, etc.)? What is the order in which the provenances are listed (from left to right) on the x-axes in Figure 2? Figure 3 - The same.
R- Figures 2, 3 and 4 were replaced, so that provenances, number of days are clear listed. Topography was not part of the current investigation, we only noted the climacteric conditions, different for each provenance.
The use of the word ‘evolution’ in lines 221-224 is unclear. R- Replaced with percentage
What are the x-axes values in Figure 4 (1-15)? Are these days? The figure legend needs to be more descriptive. R- More details were added
Are the r-values presented here p-values? Was the gemination success presented here additive? If so, wouldn’t these very significant regressions be expected (i.e., more germination over time)? If this is the case, it would be odd to test these results statistically.
R- Thank you. We have added the necessary information so that all the analyzes performed and the figures presented are clear (based on your recommendations, we have replaced Figure 4, and now is self-explanatory). In addition, we completed the title to be comprehensive, including statistics and p-value and significance.
I suggest that the information in tables be presented in a Supplment and then these results be summarized in the main body of the paper. In general, the Results section should be significantly shortened by presenting extraneous results in a supplement. The heavy results section results in a loss of focus of the study and makes the research seem overly exploratory.
R- The presentation of the obtained results in the manuscript, not as supplementary file (both thorough figures and tables) can explain the differences more clearly, in a visual manner, more easily to be followed than an enumeration of results. We consider this a more efficient way to follow the investigation and point out the most important aspects, as the data is quite copious. All tables and figures were design to have the most of the results concentrated in a structured manner, in order to respect the authors vision (seeds traits, germination treatments, seedlings growth, relationships among all these). We hope to sustain the reader to understand and read the manuscript without having to search for other files.
Discussion
The Discussion starts with a reiteration of results rather than with a discussion of the results. And this is a circumstance that is continued throughout the Discussion section. The authors should avoid repeating results in the Discussion and focus on interpreting the results within the context of the broader topic (i.e., questions asked, hypotheses addressed, application, future recommendations, etc.).
R- We tried to take into discussion the most significant results and point out the differences, causes, influences, in respect with other investigations on Robinia. The authors tested specific treatments and statistical design using data from the speciality literature, adapted to the hereby investigation. As the design and complexity of the research in new, there are no a lot of comparable data except taken individually. This is why the authors tried to put into the context the results obtained.
A citation is needed to support the assertion that the dormancy of black locust seeds is entirely physical as this was not tested in the study (lines 427-428). R- Citation was added
Given the information presented in lines 437-444 and 451-459, the novel nature of the present study is unclear. Results should be discussed such that it is clear how they expand what is already known about the species. What is primarily discussed is that the results of the present study confirm previous results. In addition, the novel nature of the present student should be described in the Introduction section.
R- Thank you for the comments, we have improved the reported aspects so that the coherence and clarity of the issues addressed have scientific coherence. Also, we hope the objectives, working hypotheses and novelty of the results obtained in the research have now a greater consistency and scientific relevance.
Reviewer 2 Report
The English grammar rules show that coma is necessary in an enumeration, before and. Please identify the situation on paper and try to fix.
Author Response
Thank you very much for the considering our work. The manuscript was verified once again and corrections were made.
Reviewer 3 Report
1) Experiment design was not clearly described, no information is available on number of replications, size of plots, randomization etc., it is recommended provide more detailed information on these elements;
2) The model of ANOVA was not provided, suggest add information on the model of ANOVA so that reader can understand how the variance was partitioned;
3) The discussion section is rather long, it is recommended divide the paragraphs into groups and add headings to the groups.
4) Make a thorough linguistic check
Author Response
1) Experiment design was not clearly described, no information is available on number of replications, size of plots, randomization etc., it is recommended provide more detailed information on these elements.
Thank you very much for the pertinent comments. Your recommendations have been helpful to us in improving the manuscript.
The seeds were collected from different providers from Romanian seed source reservations, corresponding to the Romanian Gene Reserved Forests and Seed Stands included in National Catalogue of Forest Genetic Resources and Forest Reproductive Materials. We rephrased this aspect in the Ms.
2) The model of ANOVA was not provided, suggest add information on the model of ANOVA so that reader can understand how the variance was partitioned;
R- We have filled in the text properly (we used One-way ANOVA). Thank you for the observation.
3) The discussion section is rather long, it is recommended divide the paragraphs into groups and add headings to the groups.
R- We tried to 'polish' the discussions and we revised carefully the whole text to avoid errors and some typo mistakes from the initial version of the manuscript.
4) Make a thorough linguistic check. The paper was revised.